# Life Stages and Phylogenetic Position of the New Scale-Mite of the Genus *Neopterygosoma* (Acariformes: Pterygosomatidae) from Robert’s Tree Iguana [note 1]

**DOI:** 10.3390/ani13172809

**Published:** 2023-09-04

**Authors:** Monika Fajfer, Maciej Skoracki

**Affiliations:** 1Department of Molecular Biology and Genetics, Institute of Biological Sciences, Cardinal Stefan Wyszynski University, Wóycickiego 1/3, 01-938 Warsaw, Poland; 2Department of Animal Morphology, Faculty of Biology, Adam Mickiewicz University, Uniwersytetu Poznańskiego 6, 61-614 Poznań, Poland; skoracki@amu.edu.pl

**Keywords:** scale-mites, Acari, phylogeny, ontogeny, *Liolaemus*

## Abstract

**Simple Summary:**

This research presents a description of a new ectoparasitic scale-mite species, *Neopterygosoma robertmertensi* sp. n., collected from a Robert’s tree iguana (*Liolaemus robertmertensi*) from Argentina. For the first time, the description of females was accompanied by the description of the male and juvenile stages. The morphology of all post-embryonic stages of this species was analyzed in detail using scanning electron microscopy. Additionally, we conducted a phylogenetic analysis to determine its position within the genus and created an updated identification key for all *Neopterygosoma* species. The findings show that *N. robertmertensi* sp. n. is a part of the *chilensis* group and is a sister taxon to all *Neopterygosoma* spp. collected from *Liolaemus pictus* and *L. chiliensis*.

**Abstract:**

A new pterygosomatid mite species, *Neopterygosoma robertmertensi* sp. n. (Acariformes: Pterygosomatidae) was collected from two specimens of *Liolaemus robertmertensi* (Liolaemidae) from Argentina. This new species is described based on active stages: adults (female and male) and juveniles (deutonymphs, protonymphs, and larvae) and quiescent stages (nymphchrysalis, deutochrysalis and imagochrysalis). The changes in morphological characters that occur during the ontogeny of *N. robertmertensi* have been analyzed in detail. A difference in larval sex morphology was observed for the first time in the family Pterygosomatidae (female larvae differ from male larvae in terms of the shape and size of the idiosoma and the position of the genital area). This new mite species is most similar to *N. cyanogasteri* but can be distinguished by the presence of different leg chaetotaxy patterns of genua IV and femora IV, four to six genital setae, three to five dorsomedial setae, and two to three ventromedial setae. Phylogenetic analysis was conducted based on 120 morphological characters of all *Neopterygosoma* spp. and four outgroup species using the maximum parsimony approach. The results indicated that this species is nested within mites of the *chilensis* group of *Neopterygosoma* associated with host species of the section *chiliensis* of *Liolaemus* s. str. An updated diagnosis of the *chilensis* group of *Neopterygosoma* and an identification key for all species of this genus has been provided.

## 1. Introduction

Mites of the genus *Neopterygosoma* are permanent ectoparasites, with all life stages living on the hosts. They are associated with endemic South American iguanian lizards of the genus *Liolaemus* (Sauria: Liolaemidae), and until recently, they were placed in the genus *Pterygosoma* [1,2]. The first species of this genus was described by Dittmar de la Cruz et al. [2] from tree lizards (Liolaemidae) in Argentina, exceeding the geographical range of the genus *Pterygosoma*. Later on, Fajfer and González–Acuña [1] described six new species from Chilean tree lizards and established a group *ligare* for mites associated with liolaemids. Nevertheless, the phylogenetic trees constructed by Fajfer [3] clearly showed that the genus *Pterygosoma* was paraphyletic; therefore, a new genus *Neopterygosoma* was erected for mites associated with liolaemids lizards [3]. Since then, only one new species, *N. schroederi* Fajfer, 2020, has been described [4].

Currently, mites of the genus *Neopterygosoma* are represented by eight species associated with the lizards of the genus *Liolaemus*. They are divided into two groups: *chilensis*, represented by monoxenous species associated with lizards from Chile, and *patagonica*, represented by a single oligoxenous species, *N. patagonica* (Dittmar de la Cruz, Morando and Avila, 2004), recorded on several *Liolaemus* spp. from Argentina [2,5].

Although eight species have been described in the genus *Neopterygosoma* so far, most of these descriptions are based only on a few adult females. This was necessitated by the fact that most of the described mite material was accidentally collected by herpetologists during the investigation of lizards or was taken from museum-preserved specimens, which were often washed before being fixed in formalin or alcohol. It should be emphasized that to gain a complete understanding of the mite taxonomy, phylogeny, ecology, and biology, it is essential to study both immature instars and males. In Pterygosomatidae, as in other mites, the description of juvenile stages enabled the detection of homologous features and establishment of the nomenclature used during species description [6,7]. So far, only immatures of one species, *N. schroederi*, and a male of *N. patagonica* have been described [2,4]. Nonetheless, the original description of the male was insufficient, as it only presented the idiosoma’s width and length, chaetotaxy of trochanter-tibiae I–IV, and a vague figure of the idiosoma dorsum without any details. Moreover, the type series (syntypes) consists of five males, all of which have been designated as holotypes (!), and five females. However, exact locality data were not provided; only the provinces and host species were listed separately.

In this paper, we describe a new species, *Neopterygosoma robertmertensi* sp. n., from *Liolaemus robertmertensi* from Argentina, including a first comprehensive description of the male within the genus. We extensively examine the post-embryonic stages using scanning electron microscopy, and we note differences between larval males and females for the first time within this family. Additionally, we infer the phylogenetic position of *N. robertmertensi* based on morphological data. Considering both morphology and phylogeny, this new species belongs to the *chilensis* group (the first record of Argentinian host species within the group) and is a sister taxon to Chilenian mite species associated with *Liolaemus pictus* and *L. chiliensis*. Additionally, we have revised the diagnosis of the *chilensis* group and provided an updated identification key for the genus (based on females).

## 2. Materials and Methods

### 2.1. Mite Sampling

The mite specimens were collected from the geckos housed in the herpetological collection of HUJ (abbreviations of the institutions are presented below). All lizards were kept in separate jars with 75% ethyl alcohol and were examined for mites, which were then removed from the lizards under a stereomicroscope (Nikon SMZ745 (Nikon Corporation, Tokyo, Japan). Then, the mites were placed in small vials (2 mL) containing 75% ethyl alcohol.

### 2.2. Morphological Analysis 

Before mounting in Hoyer’s medium, mite specimens were cleared and softened in Nesbitt’s solution at +45 °C for 8–48 h. All specimens were mounted as vouchers using Hoyer’s medium on a glass slide following the standard method [8].

Specimens destined for scanning electron microscopy (SEM) were dehydrated in ethanol, covered with gold, and examined using a Carl Zeiss AG–EVO^®^40 electron microscope (Carl Zeiss Microscopy, Oberkochen, Germany) at the Institute of Plant Protection of the National Research Institute in Poznan (IPP NRI), Poland. Additionally, the mites were studied and measured using a Leica DMD108 microscope (Leica Microsystems, Mannheim, Germany). All measurements, including scale bars, are given in micrometers (μm). In species descriptions, measurements (ranges) of paratypes are given in parentheses, following the data of the holotype. 

### 2.3. Terminology

In the species descriptions, names of the leg and idiosomal setae followed Grandjean [9,10], as described by Norton [6], whereas those of the palpal setae followed Grandjean [11]. Grandjean’s nomenclature [9,10] has been applied to the family Pterygosomatidae by Bochkov and O’Connor [7]. The scientific names of the lizards follow the Reptile Database [12]. All of the specimens were deposited in the arachnid collections of HUJ and CSWU. The type material of the *Neopterygosoma* spp. was loaned from the AMU. 

### 2.4. Abbreviations for Museums and Collections

AMU—Department of Animal Morphology, Adam Mickiewicz University, Poznan, Poland;

CSWU—Department of Molecular Biology and Genetics, Institute of Biological Sciences, Cardinal Stefan Wyszynski University in Warsaw, Poland; 

HUJ—National Natural History Collections of the Hebrew University of Jerusalem, Israel; 

NHM—Natural History Museum, London, the United Kingdom;

ZSM—Bavarian State Collection of Zoology, Munich, Germany.

### 2.5. Phylogeny Reconstructions Methods 

For the analysis of phylogenetic relationships between *Neopterygosoma* species, all species from the genus were used. The outgroup taxa were selected based on the analyses of Reference [3]. As a distant outgroup, *Pimeliaphilus podapolipophagus* Trägårdh, 1905 was designated, and as a close outgroup, the representatives of the genus *Geckobia* (3 spp.) of the family Pterygosomatidae were selected. We chose *G. nitidus* because it was a sister taxon to *Neopterygosoma* in the analyses of Fajfer [3], and *G. gerrhopygus* and *G. hirsti* because they were grouped separately in the analysis (see Figures 6 and 7 in Reference [3]). 

### 2.6. Cladistic Analysis

All of the characters were unordered and unweighted. In total, 13 species and 120 morphological characters of adult females were included in the analysis (Appendix A, Appendix A). Preparing and editing of the data matix were completed using NEXUS Data Editor 0.5.0 [13]. The missing states were designated as “?” and inapplicable characters as “-“. The reconstruction of phylogenetic relationships was performed in PAUP 4.0.a 147 for Microsoft Windows [14]. The branch-and-bound option was used for maximum parsimony analysis. Nodal support was evaluated using the Bremer indices calculated using PRAP2 (http://bioinfweb.info/Software/PRAP2) [15]. Analysis of the charactersdistributions and the drawing and editing of the trees were performed using FigTree v1.4.3 [16], and the final illustrations were made in Adobe Illustrator CS6.

## 3. Results

### 3.1. Systematics 

The new species described here was assigned to the *chilensis* group of the genus *Neopterygosoma* Fajfer, 2019 of the family Pterygosomatidae Oudemans, 1910, based on morphological and phylogenetic evidence. It possesses the diagnostic morphological features of the *chilensis* group (see below) and is phylogenetically nested within the *chilensis* group of *Neopterygosoma*, but with weak support (Bremer = 1).

#### 3.1.1. Description

Species group *chilensis*

Diagnosis

Body much wider (1.5–1.8 times) than long. Posteromedial part of idiosomal dorsum and venter with 3–22 pairs of dorsomedial setae or 2–21 pairs of ventromedial setae, respectively. Peripheral setae numerous and much longer than dorsal and ventral setae situated anteriorly, medially, and laterally. Setae *tc’* and *tc”* of legs II–IV serrate.

Microhabitat

Under the scales of the whole body.

Distribution and host range

This group is associated with tree lizards of the genus *Liolaemus* (Sauria: Liolaemidae) from Chile and Argentina.

Species included

*Neopterygosoma chilensis* (Fajfer and González–Acuña, 2013), *N. cyanogasteri* (Fajfer and González–Acuña, 2013), *N. formosus* (Fajfer and González–Acuña, 2013), *N. levissima* (Fajfer and González–Acuña, 2013), *N. ligare* (Fajfer and Gonzlez–Acuña, 2013), *N. ovata* (Fajfer and González–Acuña, 2013), *N. schroederi* Fajfer, 2020, *Neopterygosoma robertmertensi* sp. n.

*Neopterygosoma robertmertensi* sp. n. (Figure 1, Figure 2, Figure 3, Figure 4, Figure 5, Figure 6, Figure 7, Figure 8, Figure 9, Figure 10, Figure 11, Figure 12, Figure 13 and Figure 14).

Female (holotype, range for 14 paratypes). *Gnathosoma*. Chelicerae 145 (145–150) long. Swollen, proximal part of cheliceral base and slender distal half subequal in length, about 75 (70–75) long. Fixed cheliceral digit spinous, about 10 (10) long. Palpal femur and genu with serrate dorsal seta *dF* and *dG*, 75 (75–80) and 55 (45–60) long, respectively. Palpal tibia with slightly serrate lateral setae *l′Ti* and *l″Ti*, and with barely serrate ventral seta *vTi*. Palpal tarsi with 5 setae and solenidion (Figure 4b). Hypostome with rounded apex. Peritremes with clearly visible chambers, about 85 (85–90) long. Subcapitular seta *n* smooth or with barely discernible serration, 70–75 (75–85) long. *Idiosoma* 535 (405–550) long and 973 (715–975) wide. Dorsum (Figure 1) with an antero-mid cluster of 56 (53–60) plumose setae (20–30 long) that slightly increase in length from anterior to posterior part of this cluster. These setae are situated on smooth, weakly sclerotized propodonotal shield (Figure 2a). Laterally to this cluster about 100 (98–110) setae, 30–40 long, on each side present. About 25 (20–25) of these setae are inserted ventrally, and among them, small eyes present. Medio-lateral and postero-lateral parts of idiosoma with 48 (45–50) pairs of setae that increase in length from anterior to posterior part, 40–135 long. Dorsomedial part with 3 (3–5) pairs of serrate dorsomedial setae (*dm*). Setae *dm1* 75 (75–80) long and situated mostly anteriorly, setae *dm2*–*dm5* 90–125 (110–125) long and situated antero-laterally to the genital area. Peripheral part of body with about 30 pairs of serrate setae, 10–155 long, inserted dorsally (10–12 pairs) or ventrally (18–23 pairs). Venter (Figure 3) with 2 or 3 (2–3) pairs of serrate setae *vm*, about 80–95 long, situated laterally to genital area (Figure 2b). Genital series represented by 5 pairs of serrate setae *g1*–*g5*, 55–60 (55–60) long, 50–55 (60–65) long, 35–45 (55), 85–90 (75–95), and 70–75 long, respectively. Setae *g1*–*g4* densely serrate and situated dorsally, setae *g5* slightly serrate and situated terminally. In 3 paratypes unpaired setae *g3* present and in 5 paratypes 6 genital setae present (seta *g3* is doubled). Pseudanal setal series represented by 5 pairs of setae *ps1*–*ps5*, 75–120 long. Setae *ps1–ps3* situated terminally and *ps4*–*ps5* ventrally. *Legs.* Coxal setation *1a*, *1b*, *2b*, *3a*, *3b*, *3c* and *3d* arranged in formula 2–1–4–0. Setae *1a*, *3a*, *3b* situated outside coxal plates. All coxal setae smooth except for slightly serrate setae *3d*. Setae of trochanters I–IV: 1–1–1–1, femora I–IV: 5–4–3–2, genua I–IV: 5–4–3–2 and tibiae I–IV: 5–5–5–5. Setae *vTrI–IV*, *vFI–FIII*, *v″FI–II* filiform and smooth, *v′GI*, *v″GI–II*, *v′TiI–IV*, *v″TiI–IV*, *v′FIV*, *vGIV* with barely discernible serration, *d′FI–IV*, *d″FI–III*, *l′FI*, *d′GI–IV*, *d″GI–III*, *l′GI*, *dTiI*, *l′Ti–IV*, *l″TiI–IV* serrate. Setation of tarsi: I 14 setae (*ft*, *tc′*, *tc″*, *p′*, *p″*, *it′*, *it″*, *a′*, *a″*, *u′*, *u″*, *vs′*, *vs″*, *pl′*) and solenidion *ω1* (Figure 4a); II 10 setae (*tc′*, *tc″*, *p′*, *p″*, *a′*, *a″*, *u′*, *u″*, *vs′*, *vs″*) and *ω1*; III and IV with 10 setae each (*tc′*, *tc″*, *p′*, *p″*, *a′*, *a″*, *u′*, *u″*, *vs′*, *vs″*). Setae *tc′*, *tc″*, *it′* and *it″* of leg I represented by eupathidia; all setae *p′* and *p″* fan-like; setae *a′*, *a″*, *u′*, *u″*of legs I–IV and *tc′*, *tc″* of leg II with barely discernible serration; setae *tc′* and *tc″* of legs III–IV serrate.

Male (range for 13 paratypes). *Gnathosoma.* Chelicerae 95 long, swollen cheliceral part 40–50 long; slender distal part 45 long. Setae *dF* filiform and smooth, 50–65 long; setae *dG* filiform with barely discernible serration, 30–45 long. Supcapitular seta *n* filiform and smooth, 35–50 long. Each branch of the peritremes is about 50–70 long. *Idiosoma* 255–320 long and 435–480 wide. Dorsum (Figure 5) with barely visible propodonotal shield bearing plumose setae grouped in anterior mid-dorsal cluster (34–38 setae); these setae, 10–20 long, progressively elongate from the anterior to posterior parts of this cluster. Numerous, slightly longer plumose setae, 25–40 long, situated laterally to this cluster. Between them, small inconspicuous eyes present laterally near one long seta, about 80 long. In the medial part of the dorsum, 3 pairs of serrate setae present: *dm1–dm3* about 30–40 long, 45–65 and 60–90 long, respectively. In the lateral and posterior parts of the idiosoma, about 12 pairs of longer slightly serrate setae, 50–100 long, present, most of them situated dorsally; 2–4 pairs situated ventrally in the posterior part of the idiosoma. Aedeagus 130–140 long. Genital area with 3 pairs of setae, 5–10 long, situated on the anal valve and 3 pairs of genital papillae, 10–25 long, situated laterally to the anal valve (as in Figure 7). Venter with two pairs of ventromedial setae *vm1* and *vm2*. Setae *vm1* 40–65 long and setae *vm2* 70–75 long. *Legs*. Coxae in formula: 2–1-4–0 and all setae filiform and smooth. Setae *1a*, *3a*, *3b* outside coxal plates. Chaetotaxy of legs I–IV as in female except for lack of setae on tarsi IV. Setae *dTiI–IV*, *l′TiI–IV*, *l″TiI–IV*, *v′TiI–IV*, *v″TiI–IV*, *dGI*, *l′GI*, *l″GI*, *v′GI*, *v″GI*, *dGII*, *vGII*, *l′FII–IV*, *vFIII–IV*, *lTrI–IV* smooth; setae *l′GII*, *l″GII*, *l″FII* and *l′FIII* with barely discernible serration; setae *l′FII*, *l″FI*, *l′FIII–IV*, *dFI–III* and *vFI–II* serrate.

Imagochrysalis (tritonymph, based on 1 female and 1 male paratype). *Gnathosoma* reduced, peritremes barely visible (Figure 8b). Legs absent, only coxae I–IV visible. Idiosoma of female chrysalis (inside deutonymphal exoskeleton) 625 long and 690 wide (inside imagochrysalis fully developed coiled female with idiosoma 615 long and 685 wide present). Idiosoma of male imagochrysalis (inside larval integument) 320 long and 425 wide (inside imagochrysalis coiled fully developed male with idiosoma 295 long and 395 wide present).

Deutonymph (range for 9 paratypes). Gnathosoma as in female. Chelicerae about 90–95 long. Slender cheliceral part and swollen distal part subequal in length, about 45–50 long. Setae *dF* and *dG* slightly serrate, 55 and 40, respectively. Subcapitular setae *n* slightly serrate and 50 long. Peritremes 55 long. Idiosoma 305–330 long and 530–560 wide. Dorsum (Figure 9) with smooth propodonotal shield covered with antero-mid cluster of 26–34 setae, about 25 long. Laterally to this cluster about 26 shorter antero-lateral setae, 25–30 long, situated more anteriorly; about 30 longer antero-lateral setae, 45–60 long, situated more posteriorly; and about 10 antero-lateral short setae inserted ventrally (among them one pair of small eyes present). Dorsomedial setae represented by 3 pairs: *dm1* about 35 long, *dm2* about 50 long, and *dm3* 65 long. Peripheral setae situated dorsally (7–8 pairs) and ventrally (11–12 pairs) and about 105 long. Venter (Figure 10) with 2 ventromedial setae *vm1* and *vm2.* Genital region with 3 setae *g1–g3.* Setae *g1* and *g2* 20–25 long, setae *g3* 35–45 long. Pseudanal setal series represented by 3 pairs of setae *ps1–ps3*, 70–75 long. Legs as in female, except for lack of setae *vTrIV*.

Deutochrysalis (based on 2 paratypes in exoskeleton of protonymph). Gnathosoma reduced, with barely discernible peritremes. Idiosoma 415–360 long and 620–650 wide. Legs absent, only coxae I–IV present. Inside deutochrysalis fully developed deutonymph present.

Protonymph (range for 5 paratypes). Gnathosoma. Chelicerae 95 long; slender cheliceral part and swollen distal part subequal in length, 45–50 long. Hypostome with rounded apex. Setae *dF* and *dG* slightly serrate, 40–60 and 40–45 long, respectively. Subcapitular seta *n* filiform and smooth, about 50 long. Each branch of peritremes about 60 long. Idiosoma 315–345 long and 535–550 wide. Dorsum (Figure 11) with weakly sclerotized propodonotal shield with densely plumose setae grouped in anterior mid-dorsal cluster (27–42 setae). These setae subequal in length, 20–25 long. Numerous (about 63–67 pairs) of slightly longer plumose setae, 25–40 long, situated laterally to this cluster. Between them small inconspicuous eyes present. In the medial part, 3 pairs of setae *dm1* (30 long), *dm2* (55–65) and *dm3* (60–70) present, and about 20–28 pairs of postero-lateral setae, 40–95 long. Venter (Figure 12) with setae *vm1*, 55 long, and about 29 pairs of serrate peripheral setae in postero-lateral part of the idiosoma, 60–70 long. These setae situated: ventrally (12 pairs), terminally (7–8 pairs), and dorsally (10–11 pairs). Genital area with 3 pairs of genital setae *g1*–*g3* 30, 15, and 25 long, respectively; and with 3 pairs of densely serrate pseudanal setae *ps1–3*, 70–80 long. Legs. Coxal setae *1a*, *1b*, *2b*, *3a*, *3b*, *3c* filiform and smooth, setae *3d* slightly serrate. Setae *1a* and *3a* situated outside coxal plates. Chaetotaxy pattern of legs I–IV as in female, except for lack of setae *vTrIV*.

Nymphchrysalis (based on two specimens in larval exoskeleton). Gnathosoma reduced, with barely discernible peritremes. Idiosoma 225–240 long and 350–360 wide with completely developed protonymph inside, about 205 long and 330 wide. Legs absent, only coxae I–IV visible. 

Larva (range for 8 larval male paratypes and 11 larval female paratypes). Gnathosoma. Chelicerae about 50 long; swollen cheliceral part 20–25 long and slender distal part about 30 long. Fixed cheliceral digit absent. Tarsi with 5 setae and solenidion (Figure 14b). Each part of peritremal branch 35–40 long. Setae *dG* 20–40 long, setae *dF* 40–50 long. Subcapitular setae *n* absent. Idiosoma wider (290–360 wide) than long (170– 250) in female larvae and almost as long as wide in male larvae (155–200 long and 170–215 wide). Dorsum without propodonotal shield (Figure 14a) and with 11 plumose setae situated as in Figure 13a,c. Five setae situated in anterior part thicker and shorter (15–30 long) than narrower and longer (35–50 long) setae situated in posterior half of idiosoma. Eyes present on lateral margins of idiosoma. Venter (Figure 13b,d) devoid of any setation. Genital area (Figure 14d) with three genital setae *g1*–*g3*, 10–15 long and two pseudanal setae *ps1* and *ps2*. Setae *ps1* 40–50 long and *ps2* 30–50 long. Legs. Coxae in formula: 2–0–1; setae *1a*, *1b*, *3a* filiform and smooth. Chaetotaxy of legs I–IV as follows: (5–5–5) (2–2–1) (4–4–3) (0–0–0). Setae *dTiI–III*, *l′TiI–III*, *l″TiI–III*, *vTiI–III*, *dl′GI–III*, *dl″GI–II*, *dl′FI–III*, *dl″FI–III* filiform and slightly serrate. Setae *vFI–II* with barely discernible serration and setae *dFI–III* serrate. Setation of tarsi: I 11 setae (*ft*, *p′*, *p″*, *it′*, *a′*, *a″*, *u′*, *u″*, *vs′*, *vs″*, *pl′*) and solenidion *ω1*; II 10 setae (*tc′*, *tc″*, *p′*, *p″*, *a′*, *a″*, *u′*, *u″*, *vs′*, *vs″*) and *ω1*; III and IV with 10 setae each (*tc′*, *tc″*, *p′*, *p″*, *a′*, *a″*, *u′*, *u″*, *vs′*, *vs″*). Setae *vs′*, *vs″*, *a′*, *a″*, *pl′* smooth or with barely discernible serration, setae *p′* and *p″* fan-like, setae *tc′*, *tc″* of legs II–III slightly serrate (*tc’* longer than *tc*”), setae *ft* smooth, setae *it’* in form of eupathidion (Figure 14c). 

Eggs 170–180 long 150–160 wide.

Type material

Holotype and 8 female, 12 male, 9 deutonymph, 4 protonymph, 2 imagochrysalis, 1 deutochrysalis, 1 nymphchrysalis, 8 male larvae, and 10 female larvae paratypes from *Liolaemus robertmertensi* Hellmich, 1964 (HUJ no. 17923) (Iguania: Liolaemidae), Argentina, Catamarca, 30 km south of Andalgalá, September 1987, coll. O. Pagaburo and Yehudah L. Werner; 7 female, 1 male, 1 deutonymph, 1 nymph chrysalis, 1 protonymph chrysalis, 1 dutonymph chrysalis, and 1 female larva paratypes from same host (HUJ no. 18091) and location, September 1987, coll. O. Pagaburo and Yehudah L. Werner. 

Type Material Deposition

Female holotype and most paratypes were deposited in the HUJ (reg. HUJINV-Acari_Pte00003.1–38 and HUJINV-Acari_Pte00004.1–11), except for six female, three male, three deutonymph, two protonymph, and four larvae paratypes in the CSWU (reg. no. CSWU-Pte20.1.1–16 and Pte20.2.1–2).

Etymology

The species name is derived from the species name of the host.

Differential diagnosis

This species is most similar to *Neopterygosoma cyanogasteri* from *Liolaemus cyanogaster* (Duméril and Bibron) from Chile [1]. In females of both species, the setation of tarsi I–IV, tibiae I–IV, genua I–III, femora I and III, and trochanters I–IV is the same, fixed cheliceral digit is spinous, palp seta *dF* is longer than *dG*, subcapitular seta *n* is smooth or with barely discernible serration, the antero-mid cluster of dorsal setae is represented by about 60 setae, and five pseudanal setae *ps* are present. In *Neopterygosoma robertmertensi* setae *lv’GIV*, *lv’GII* and *ld’FIV* are absent, coxal setae *3a* are smooth, 4–6 pairs of serrate genital setae are present, 3–5 pairs of dorsomedial setae, and 2 or 3 pairs of ventromedial setae are present. In *N. cyanogasteri* setae *lv’GIV*, *lv’GII* and *ld’FIV* are present, coxal setae *3a* are serrate, one smooth genital seta, 17–21 dorsomedial setae, and 14–18 ventromedial setae are present.

Remarks

Our research used scanning electron microscopy to enhance taxonomic descriptions of the new *Neopterygosoma* species. As a result, we noticed that in the original description of *Neopterygosoma* spp. [1], some inaccuracies are mentioned. The detailed photographs revealed that a smooth and weakly sclerotized propodonotal shield is present in all *Neopterygosoma* mites (Figure 4b) (it appears in protonymph).

#### 3.1.2. Key to species of Neopterygosoma (Females) (Based on the Key of Fajfer [4]) 

1.Body much wider than long (1.5–1.8 times). Setae *tc’* and *tc”* of legs II–IV serrate. Peripheral setae much longer than dorsal and ventral setae situated anteriorly, medially and laterally…*chilensis* group 2-Body circular, only slightly wider than long (1.1–1.3 times). Setae tc’ and tc” of legs II–IV smooth. Peripheral setae subequal with anterior, medial and lateral setae on idiosomal dorsum and venter…patagonica group…*N. patagonica* (Dittmar de la Cruz, Morando and Avila, 2004)2.Five setae on genu I and 5 pseudanal setae *ps*…3-Four setae on genu I and 3 pseudanal setae *ps*…*N. formosus* (Fajfer and González–Acuña, 2013)3.Four setae on femur II…4-Five setae on femur II…54.Five pseudanal setae present. Setae *vTrI–IV* densely serrate. Swollen cheliceral part of chelicerae shorter than slender distal part. Subcapitular setae *n* short (45–65 long)…*N. chilensis* (Fajfer and González–Acuña, 2013)-Four pseudanal setae present. Setae *vTrI–IV* smooth. Swollen cheliceral part of chelicerae longer than slender distal part. Subcapitular setae *n* long (about 125 long)… *N. schroederi* Fajfer, 20195.Three setae on femur IV. One pair of genital setae *g1*. Dorsomedial setae *dm* represented by 15–21 pairs of setae. Ventro–medial setae *vm* represented by 10–18 pairs…6-Two setae on femur IV. Four or five pairs of genital setae. Dorsomedial setae *dm* represented by 3–5 pairs of setae. Ventromedial setae *vm* represented by 1–3 pairs…*N. robertmertensi* sp. n.6.Genital setae smooth. Fixed cheliceral digit spinous, palp setae *dF* serrate only distally, subcapitular setae *n* serrate…*N. cyanogasteri* (Fajfer and González-Acuña, 2013)-Genital setae serrate. Fixed cheliceral digit reduced to rounded structure, palp setae *dF* serrate on all length, subcapitular setae *n* smooth…77.Coxal fields I with 2 setae. Gnathosoma situated apically. Free peritremal branch present. Setae *dG* serrate on all length…8-Coxal fields I with 3 setae. Gnathosoma displaced on dorsal side. Free peritremal branch absent. Setae *dG* serrate only at distal tip…*N. ovata* (Fajfer and González-Acuña, 2013)8.Antero-medial setae increase in length from anterior to posterior part of setal cluster. Setae *a’* and *a”* of tarsi I slightly serrate. Setae *v’TrI–IV* serrate. Setae *3a* smooth and situated outside coxal plates…*N. levissima* (Fajfer and González–Acuña, 2013)-Antero-medial setae subequal in length. Setae *a’* and a” of tarsi I smooth. Setae *v’TrI–IV* with barely discernible serration. Setae *3a* slightly serrate and situated on coxal plates…*N. ligare* (Fajfer and González-Acuña, 2013).

### 3.2. Phylogeny

#### Unweighted Parsimony Analysis

The analysis of the data matrix (Appendix A) showed that out of 120 characters (Appendix A), 85 were informative. The analysis with all characters treated as unordered and unweighted was performed with Paup and produced one parsimonious tree (Figure 15). The tree is 219 steps long and has a consistency index (CI) of 0.64; retention index (RI) of 0.56, and rescaled consistency index (RC) of 0.36. 

The monophyly of the genus *Neopterygosoma* is supported by four synapomorphies (Bremer index 3), of which two are unique (length of coxae I, absence of coxal setae *2a* and *4a*). As expected, the resulting topology in this analysis is very similar to that in Fajfer [3]; in that hypothesis, the *P. patagonica* was the sister taxon to three species of Chilean species (*P. chilensis*, *P. ligare*, and *P. formosus*) included in the analysis. Our analysis confirms that *N. patagonica* from Argentina, considered less specialized (it has a circular body shape that is unable to hide under the host’s scales), is the sister group to all the other species of the genus from Chile, considered more specialized (their idiosoma is wider than long, therefore, they live completely hidden beneath the scales). Its position is also supported by five common synapomorphies (Bremer index 2), of which three are unique (e.g., the presence of much longer setae in the postero-lateral part and peripheral part of idiosoma than in the anterior half of the dorsum).

The new species, *L. robertmertensi*, is a sister taxon to all species collected from *Liolaemus pictus* (*N. formosus*, *N. ovata*, *N. ligare*, and *N. levissima*) and *L. chiliensis* (*N. chilensis*) and is supported by the presence of five non-unique synapomorphies (Bremer index 1). The node uniting all of the above-mentioned mite species collected from the two host species is supported by five non-unique synapomorphies (Bremer index 2). Within the clade, the relationship within the species is weakly supported: *N. formosus* is a sister taxon to *N. ovata + N. ligare* (Bremer index 1), and the three species are a sister group to *N. levissima* + *N. chilensis* (Bremer index 1). Notably, the positions of both *N. schroederi* and *N. cyanogasteri*, are weakly supported by several non-unique synapomorphies (Bremer index of 1).

The only differences between the tree presented in Reference [3] and this tree lay in the position of the outgroup species. In the analysis [3], the genus *Geckobia* was paraphyletic with *G. nitidus* as a sister taxon to representatives of species of the genus *Neopterygosoma*, while *Geckobia gerrhopygus* + *G. hirsti* were as a sister taxon to the genera: *Gerrhosaurobia* + *Zanurobia* + *Ixodiderma* + *Scaphotrix* + *Pterygosoma*. In our analysis, all the outgroup *Geckobia* spp. are grouped in a common clade.

## 4. Discussion

The genus *Liolaemus* is the most ecologically diverse and species-rich genus distributed in South America from the high Andes of central Perú to the shores of Tierra del Fuego, and it spans an altitudinal range from sea level to over 5000 m [17]. The liolaemid lizards cover various climatic regimes and inhabit a great diversity of habitats (e.g., arid Atacama desert or humid rainforests). Moreover, the lizards exhibit a wide range of reproductive modes, types of diets, coloration patterns, and body sizes [18]. They also have a long evolutionary history dating back to 18–22 million years ago [19,20]. 

Currently, the genus includes over 280 species [12], but new species are being discovered at a rapid rate every year, e.g., [21,22]; therefore, it is estimated that the actual number of the species may be much higher. The genus is subdivided into two subgenera—*Liolaemus* (sensu stricto) and *Eulaemus* [23]—which appear to have separated at least 12.6 million years ago and are currently each divided into several groups. The presence of *Neopterygosoma* mites has been detected in 12 different species of hosts belonging to *Liolaemus* s. str. living on both sides of the Andes at different elevations, having different types of scales, coloration patterns, etc. [18,24]. 

As a rule, mites from different pterygosomatid genera are strictly specific with respect to lizard hosts, and cospeciation has a strong influence on the architecture of host–parasite relationships within the family Pterygosomatidae [3]. All representatives of the genus *Neopterygosoma* are monoxenous parasites (the *chilensis* group) except for *N. patagonica* collected from several *Liolaemus* spp. (oligoxenous parasite). Since host species from the same communities (these host taxa distributions partially overlap [17]) do not carry the same set of parasite species, we can expect to observe at least partially parallel evolution of *Neopterygosoma* mites of the *chilensis* group and *Liolaemus* hosts. 

Nonetheless, the co-phylogenetic studies require phylogenetic hypotheses or data matrices for both lineages involved in the coevolutionary process. So far, the relationships between *Liolaemus* lizards at the species level are still questionable, e.g., [19,25]. Recently, Troncoso–Palacios et al. [26] conducted a phylogenetic study of the relationship of species of *Liolaemus* s. str. (based on three fragments of the mitochondrial genome); as a result, the species were divided into two main clades named: *chiliensis* and *nigromaculatus* sections. These findings were congruent with the phylogenetical tree (Figure 3 in Reference [17]) based on previous works [19,25,27]. Until now, all *Neopterygosoma* spp. are associated with closely related hosts belonging to the *chiliensis* section, whereas representatives of another pterygosomatid genus, i.e., *Geckobia nitidus* and *G. zapallarensis*, were collected from lizards of the *nigromaculatus* section [28] (marked on Figure 15). 

However, not all of the host species groups were recovered monophyletic in the work of Troncoso–Palacios et al. [26]; therefore, Parenza et al. [29] infer a robust phylogeny (based on 541 ultra-conserved elements and 44 protein-coding genes) for a Chilenian clade of *Liolaemus* s. str. using representatives of all thirteen groups. As a result, only the relationship among the major Chilean clade of *Liolaemus* was resolved, as in previous studies [26] (Figure 15). All mites of the *chilensis* group (i.e., monoxenous ‘more specialized’ mite species) have been associated with closely related hosts belonging to three host groups of [26], i.e., *robertmertensi*, *gravenhorsti*, and *pictus*. The pterygosomatids have been found on all representatives within the two former groups except for *L. sanjuanensis* (*robertmertensi* group) and *L. gravenhorsti* (*gravenhorsti* group), which suggests that checking numerous host specimens of the two species for mites might lead to new species descriptions. 

The highest number of *Neopterygosoma* spp. was described from a single host species—*L. pictus* (4 spp.)—belonging to the *pictus* group, including 11 host species. However, the number of species in this group is debatable because a few species have been treated as subspecies of *L. pictus* [30,31] or synonymized with *L. pictus* [32]. This host species has a wide distribution and forms a local population at low elevations (0–1600 a.s.l.) on both sides of the Andes, whereas the remaining *Liolaemus* spp. live either in the eastern or western slopes of the mountains [19]. It is unknown if the mite species occupy the full geographical range of their main host because so far, they have been found only in Isla Mocha (Arauco Province, Chile), although attempts to collect the mites from different localities were made (by M. Fajfer in ZSM and NHM). This could be interpreted as a consequence of the recent evolution of new mite species which are competing on the same host; therefore, further studies may prove that this group of parasites undergoes rapid adaptive radiation.

Our phylogenetic analysis shows that *N. patagonica* is a sister taxon to all monoxenous mites of the *chilensis* group. It agrees with the findings of Fajfer [3]. *P. patagonica* inhabits various host species of three different groups (see Figure 15) [17,26], which might suggest that this mite species’ association is not fully recovered, and we can expect even more multi-host associations. *P. patagonica,* due to its circular shape of idiosoma, is morphologically unable to take shelter under the scales; therefore, most of its idiosoma protrudes beyond the scales. This probably allows the mite, by virtue of its effective dispersal abilities, to switch off quickly from a host when the opportunity arises, and then locate and colonize another host. This is especially probable if the host species, as in this case, share the same diet and occur at least partially in the same habitat [17]. 

The phylogenetical analysis indicates that the newly described species, *N. robertmertensi*, is nested within the mites of the *chilensis* group of *Neopterygosoma* associated with species of the section *chiliensis* of *Liolaemus* s. str. Its placement is also confirmed by a set of morphological features (see Figure 15), although the Bremmer support is only 1. The reason for this may be that *N. robertmertensi* has many unique features (e.g., the number of dorsomedial, ventromedial, or genital setae, i.e., characters 36, 40, and 49–52 in Figure 15, respectively), which do not match the description of the *chilensis* group provided in Reference [4]. Therefore, a revised description of the species group is presented here.

For the first time, we collected all mites from the host specimens that were preserved directly after collection. As a result, we collected hundreds of mites which were completely hidden beneath the lizard’s scales. We found 1–12 specimens under a single scale, and the mites inhabited each body part of the host specimens. This lack of topical (habitat) specificity is quite surprising because in pterygosomatids living under the scales (such as *Pterygosoma* or *Geckobia*), a high preference towards a microhabitat on the host body is observed [33,34]. 

This large number of mites allowed us to observe morphological diversity among juveniles and adults and to illustrate for the first time the complete morphological ontogeny of these mites. For the first time in the family Pterygosomatidae, we were able to determine differences between the sexes of larvae. In male larvae of *N. robertmertensi*, the idiosoma is smaller and almost as long as wide (155–200 long and 170–215 wide), the genital region is situated ventrally, and the male develops directly in chrysalis inside the larval integument. In female larvae, the idiosoma is bigger and wider than long (170–250 long and 290–360 wide), the genital region is situated terminally, and the life cycle of the larva consists of both: active stages that feed on blood (protonymph, deutonymph, and adult) and legless inactive stages (nymphchrysalis, deutochrysalis, and imagochrysalis). 

Our study shows that a female larva forms a chrysalis that resembles those found in other pterygosomatids (e.g., see Figure 8C in Reference [35]). Inside the chrysalis, a coiled protonymph develops. After molting, the newly emerged protonymph is larger than larva, and we observe the appearance of four pair of legs with the full set of setae on femora–tarsi IV, numerous idiosomal setae arranged similarly to subsequent stages, subcapitular setae *n*, weakly sclerotized small propodonotal shield, additional setae *ps3* in the genital region, leg setae on coxae II–III (*2b*, *3b*, *3c*, *3d*), genua-trochanter I–III (*v′GI–III*, *v″G–III*, *l′GI*, *vFI*, *vTrI–III)*, and tarsi I, i.e., *it″* (ζ), *tc′* (ζ) and *tc*″ (ζ). 

In the protonymph integument, we observed a deutochrysalis with a completely formed coiled deutonymph. This stage differs from a protonymph by the presence of much smaller gnathosoma and longer palpal setae (*dF*, *dG*), fewer setae on the mid-dorsal cluster, and the arrangement of setae (fewer in number) that resembles that in females. An adult female develops in the imagochrysalis (tritonymph). It differs from a deutonymph by the size of the idiosoma, the presence of additional two or three genital setae (*g4*–*g6*) and pseudanal setae (*ps4*–*ps5*), and ventral setae on trochanter IV. The males develop directly in the chrysalis inside the exoskeleton of larvae. 

At this point, it is unclear whether the presence of both male and female larvae is unique for the genus *Neopterygosoma*. In Pterygosomatidae, as a rule, the description of juvenile morphology is often neglected. This could be due to several factors, such as (i) the difficulty of associating juveniles with an adult if the adults are missing in the sample, (ii) a small number of specimens found on hosts in museum collections (the mites might fall off the host during its preservation), (iii) the presence of only female mites on hosts, which may be explained by the short duration of their juvenile stages or (iv) the small size and transparency of the juvenile stages which make them difficult to notice on the hosts.

It is interesting to note that the larvae of *Neopterygosoma* differ from those of other genera, such as *Pterygosoma* or *Geckobia*, due to the absence of setae on tarsi I, specifically *it″* (ζ), *tc′* (ζ) and *tc″* (ζ). In other pterygosomatid larvae, only one fan-like proral setae *p′*, one simple tectal seta *tc′*, and paired iterals *it′* and *it″* in the form of eupathidia are present. Additionally, Norton’s description of leg chaetotaxy [6], based on Grandjean’s work [10,11], referred to the iterals as “post-larval setae” that are added in the protonymph stage. Yet, in *Neopterygosoma* spp. larvae, there is only one euphatidial setae *it’* while in contrast, the larvae of *Pterygosoma* have a pair of iterals (*it’* and *it*”).

## 5. Conclusions

In this research, we meticulously described and illustrated the morphology of the new species of pterygosomatid mite, *Neopterygosoma robertmertensi*, using scanning electron microphotography. As a result, we found new morphological features which were not recognized in previous studies of *Neopterygosoma* spp., such as the presence of a weakly sclerotized propodonotal shield. We observed the species morphological ontogeny and analyzed the main morphological differences between juvenile stages. For the first time in Pterygosomatidae, we observed both male and female larvae that differ mainly by the size and shape of idiosoma and from other pterygosomatid larvae by chaetotaxy of tarsi I. Additionally, the phylogenetic analysis showed that this species is nested within the *chilensis* group of *Neopterygosoma*, which was consistent with the morphological analysis. *Neopterygosoma* mites occur only on hosts belonging to three groups of the *chiliensis* section of the subgenus *Liolaemus* s. str., whose distributions partially overlap. Nonetheless, the hosts do not carry the same sets of parasite species. This suggests that mites of the *chilensis* group might be a good fit for cophylogenetic studies, especially if we take into account the fact that some studies conducted on pterygosomatid mites revealed a cophylogenetic pattern [3].

## Figures and Tables

**Figure 1 animals-13-02809-f001:**
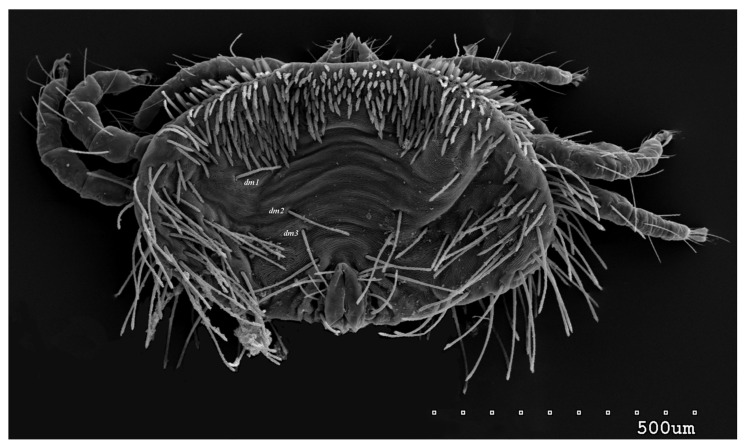
*Neopterygosoma robertmertensi* sp. n., female in dorsal view.

**Figure 2 animals-13-02809-f002:**
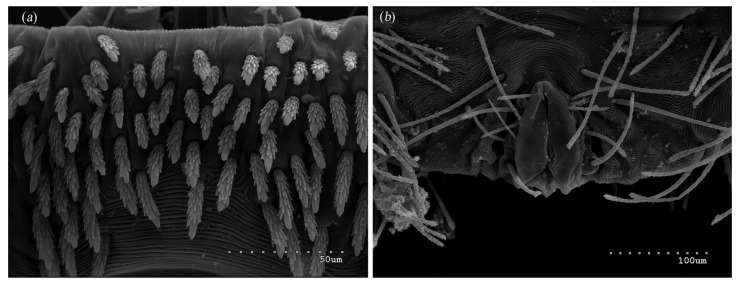
*Neopterygosoma robertmertensi* sp. n., female details: (**a**) propodonotal shield (**b**) genital region.

**Figure 3 animals-13-02809-f003:**
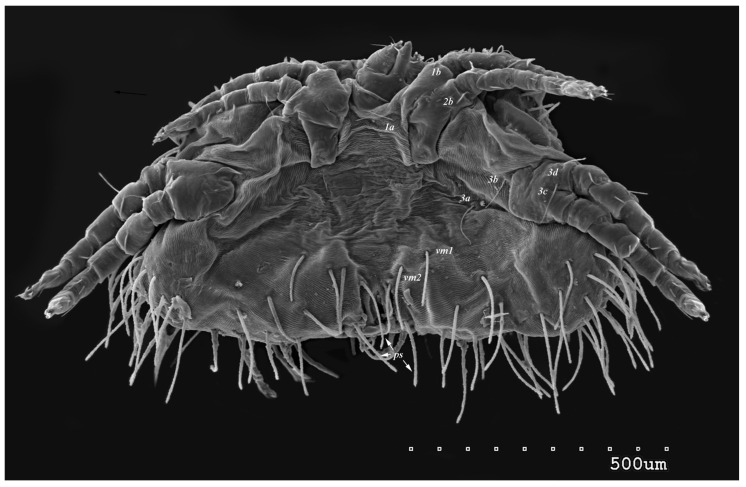
*Neopterygosoma robertmertensi* sp. n., female in ventral view.

**Figure 4 animals-13-02809-f004:**
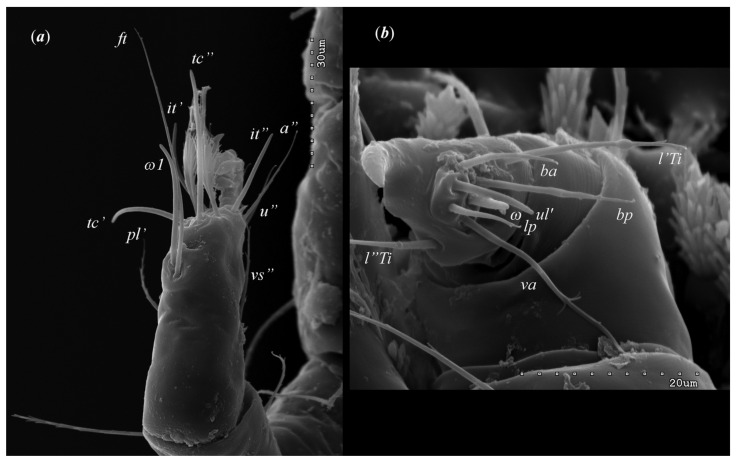
*Neopterygosoma robertmertensi* sp. n., female details: (**a**) tarsi I in dorsal view; (**b**) palps in ventral view.

**Figure 5 animals-13-02809-f005:**
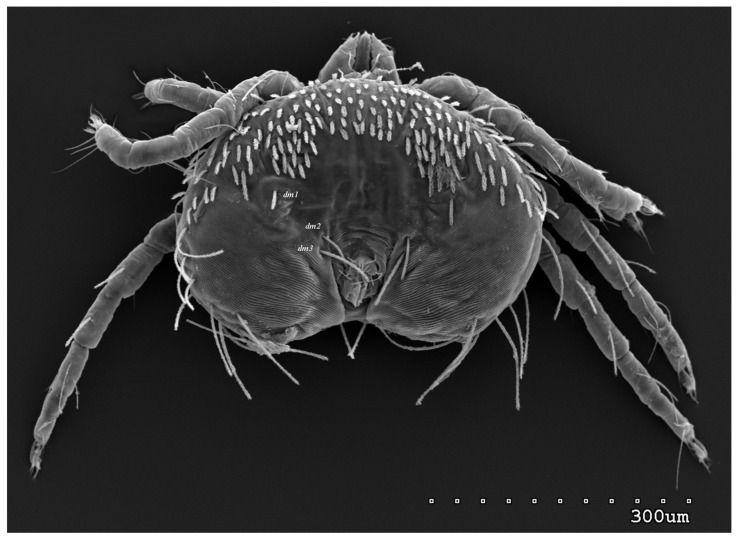
*Neopterygosoma robertmertensi* sp. n., male in dorsal view.

**Figure 6 animals-13-02809-f006:**
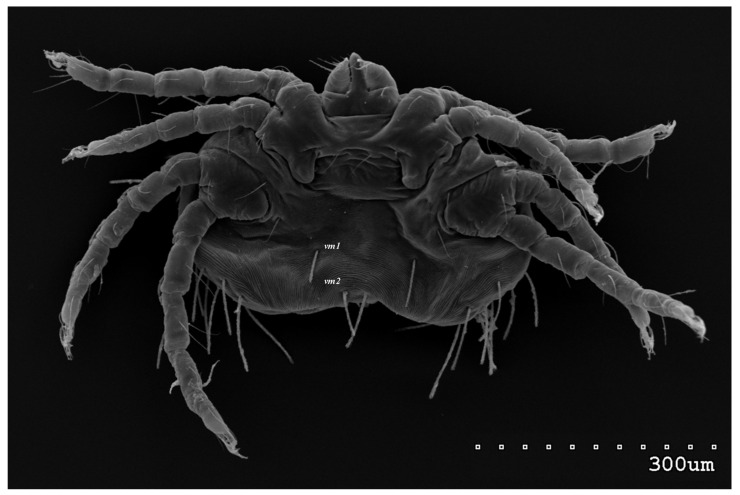
*Neopterygosoma robertmertensi* sp. n., male in ventral view.

**Figure 7 animals-13-02809-f007:**
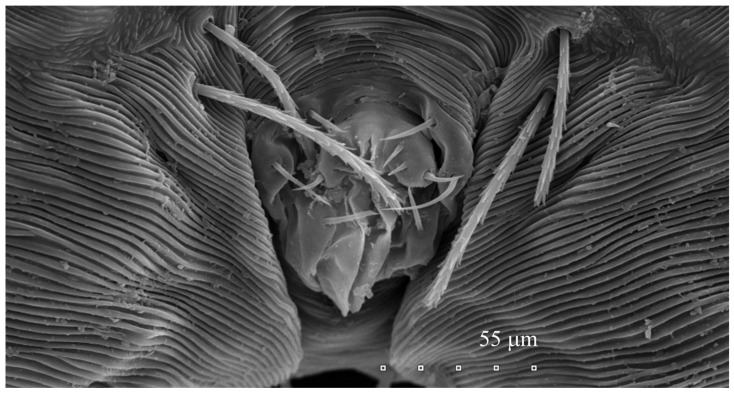
*Neopterygosoma robertmertensi* sp. n., male, genital area, enlarged.

**Figure 8 animals-13-02809-f008:**
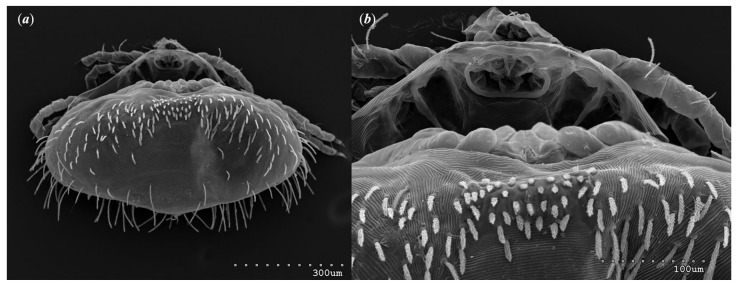
*Neopterygosoma robertmertensi* sp. n. (**a**) imagochrysalis in the exoskeleton of deutonymph, dorsal view; (**b**) reduced gnathosoma, peritremes and coxae I–II of imagochrysalis, enlarged.

**Figure 9 animals-13-02809-f009:**
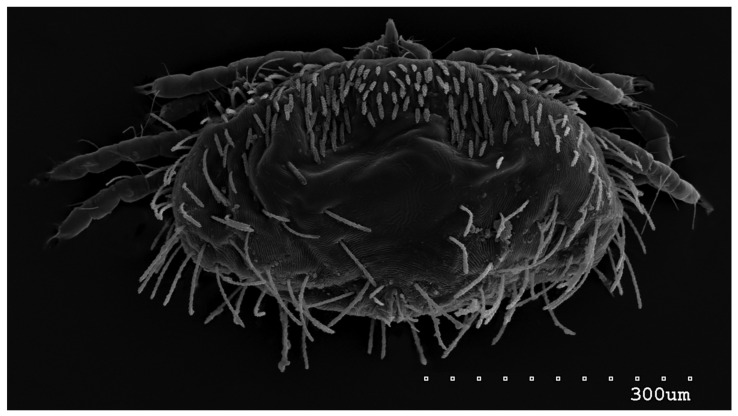
*Neopterygosoma robertmertensi* sp. n., deutonymph in dorsal view.

**Figure 10 animals-13-02809-f010:**
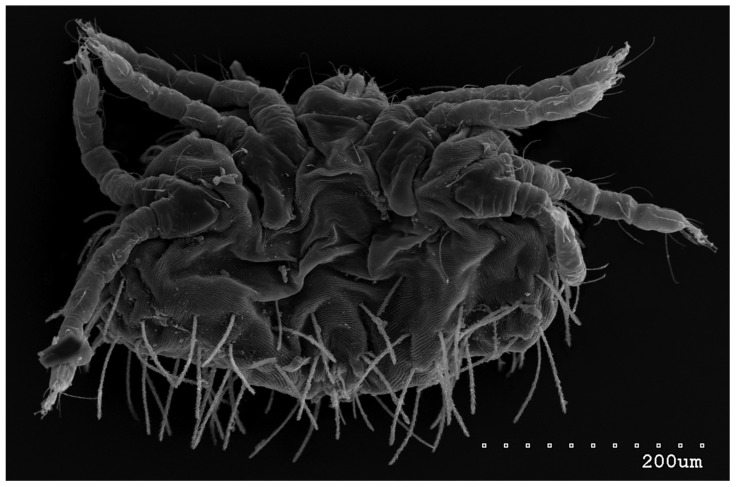
*Neopterygosoma robertmertensi* sp. n., deutonymph in ventral view.

**Figure 11 animals-13-02809-f011:**
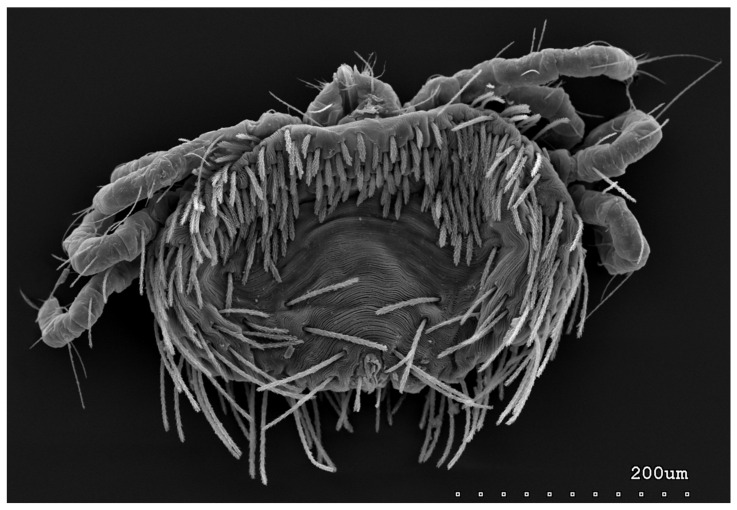
*Neopterygosoma robertmertensi* sp. n., protonymph in dorsal view.

**Figure 12 animals-13-02809-f012:**
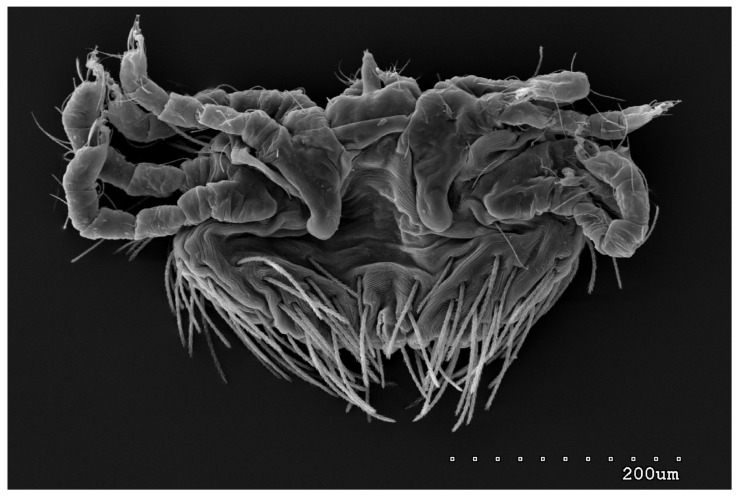
*Neopterygosoma robertmertensi* sp. n., protonymph in ventral view.

**Figure 13 animals-13-02809-f013:**
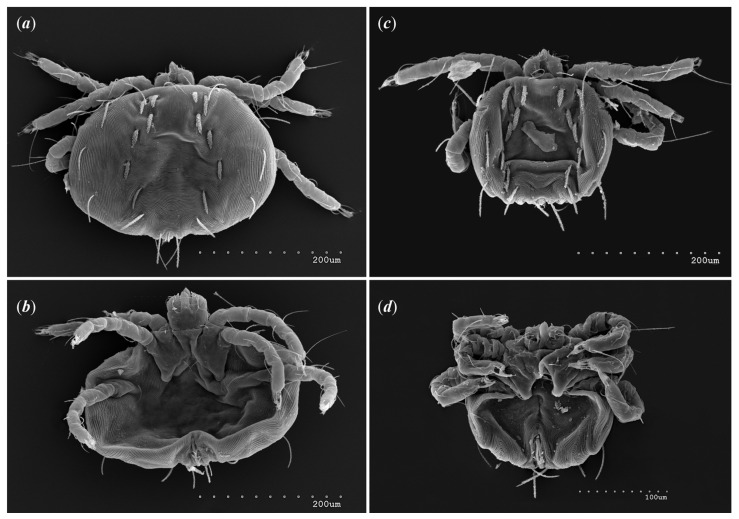
*Neopterygosoma robertmertensi* sp. n. (**a**) female larva in dorsal view; (**b**) female larva in ventral view; (**c**) male larva in dorsal view; (**d**) male larva in ventral view.

**Figure 14 animals-13-02809-f014:**
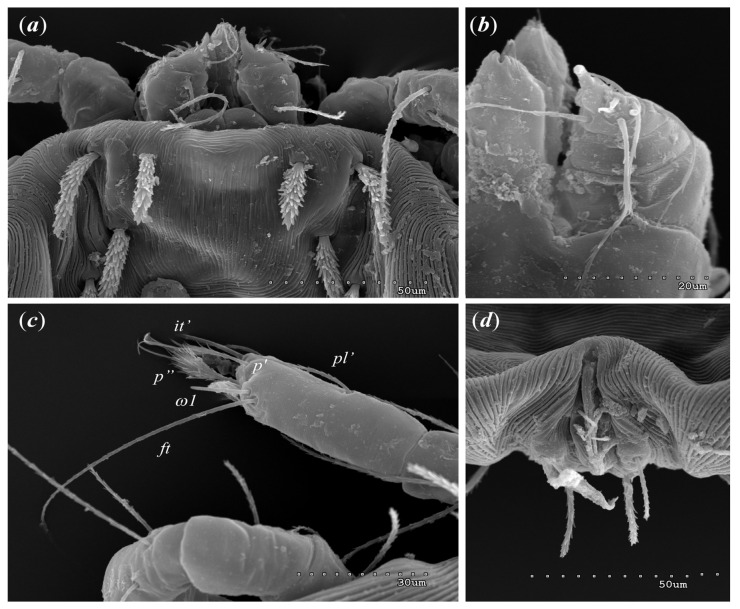
*Neopterygosoma robertmertensi* sp. n. larva, details (**a**) dorsal setae; (**b**) part of gnathosoma in ventral view; (**c**) tarsi I in dorso-lateral view; (**d**) genital region.

**Figure 15 animals-13-02809-f015:**
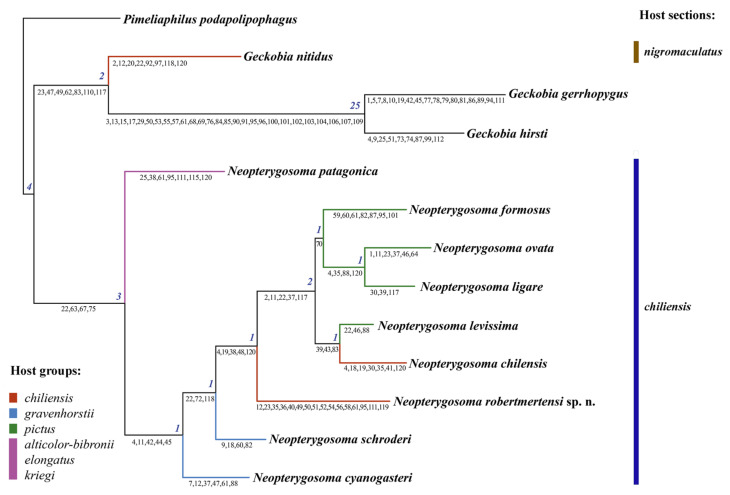
The most parsimonious tree (tree length 219, CI of 0.64, RI of 0.56, RC of 0.36) found using the branch-and-bound search option for the unordered and unweighted dataset. Numbers at nodes are Bremer indices. Numbers below branches are common synapomorphies (character numbers refer to Appendix A). Distribution of the mite species within host groups and section are marked in different colours.

## Data Availability

The material is stored in Cardinal Stefan Wyszynski University (Warsaw, Poland) and will be shared upon reasonable request to Monika Fajfer.

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
