# Peer review of "Life Stages and Phylogenetic Position of the New Scale-Mite of the Genus Neopterygosoma (Acariformes: Pterygosomatidae) from Robert’s Tree Iguanaâ€"

_animals, 2023, doi:10.3390/ani13172809_

Round 1
Reviewer 1 Report
The presented work is a meticulous and appropriate description of the new species of ectoparasite N. robertmertensi, there are many interesting results such as: the sexual dimorphism noticed in the larvae and reflections on post embryonic development. Very interesting phylogenetic tree results, acceptable disimilarities of the branches from previous work.
In my opinion the structure of the experiment and the quality of the work deserve to be accepted.
Author Response
Dear reviewer, thank you very much for your review.
Reviewer 2 Report
Dear Authors, congratulations on the excellent taxonomic paper!
Please find my remarks below:
line 38: Please specify "premanent parasites" - does it mean that all the instars never leave the host?
lines 58-60: "However, it should be emphasized that to gain a complete understanding of the mite taxonomy, phylogeny, ecology, and biology, it is essential to study both immature instars and males" - I suggest that however is not at the beggining of sentence: " It should be emphasized, however,...." The same for line 63..
Author Response
Dear reviewer, thank you very much for all your suggestions. They were incorporated into the body of the manuscript.
Reviewer 3 Report
Dear Authors,
This manuscript presents the description of a new scale-mite species of the genus Neopterygosoma collected from Robert's Tree Iguana.
The knowledge about these mites is limited so the discovery of new species fills this knowledge gap. It is very important that all postembryonic stages are described for the first time within the family and the validity of the new species is strengthened by the phylogenetic analysis conducted. The descriptions provided are very detailed and the SEM photos are of excellent quality.
I have no corrections for the text but I strongly suggest the addition of some line drawings depicting some body parts that are important for identification and are not clealy visible from the provided photos. These drawings can be easily used from other authorities for comparison. I suggest the addition of detailed drawings of gnathosoma (subcapitulum, chelicerae and palps) and legs, at least for the holotype female.
Sincerely
Author Response
Dear reviewer, thank you very much for all your suggestions. I did not include the line drawings of gnathosoma and leg tarsi in the manuscript because the SEM figures are of good quality therefore the setae are named on the SEM figures. Moreover, the arrangement of the setae on gnathosoma and leg tarsi are the same within the genus Neopterygosoma. Therefore, if the reader has a problem with identification may compare it with line drawings presented in other publications such as i.e. with the details in Fajfer 2020 (e.g. on Figure 3 and 5)
Round 2
Reviewer 3 Report
I recommend the acceptance of the manuscript in its present form